# Capturing GEI Patterns for Quality Traits in Biparental Wheat Populations

**Ivana Plavšin** [1,2] , **Jerko Gunjača** [2,3,*] , **Ruđer Šimek** [1,†] **and Dario Novoselović** [1,2]

1   Department for Cereal Breeding and Genetics, Agricultural Institute Osijek, Južno Predgrađe 17, 31 000 Osijek, Croatia; ivana.plavsin@poljinos.hr (I.P.); rudjer.simek@gmail.com (R.Š.); dario.novoselovic@poljinos.hr (D.N.)
2   Centre of Excellence for Biodiversity and Molecular Plant Breeding (CoE CroP-BioDiv), Svetošimunska Cesta 25, 10 000 Zagreb, Croatia
3   Department of Plant Breeding, Genetics and Biometrics, University of Zagreb Faculty of Agriculture, Svetošimunska Cesta 25, 10 000 Zagreb, Croatia
*   Correspondence: jgunjaca@agr.hr; Tel.: +385-1-239-3938
†   Current address: Arguo d.o.o., Zadarska 14, 31 000 Osijek, Croatia.

**Abstract:** Genotype-by-environment interaction (GEI) is often a great challenge for breeders since it makes the selection of stable or superior genotypes more difficult. In order to reduce drawbacks caused by GEI and make the selection for wheat quality more effective, it is important to properly assess the effects of genotype, environment, and GEI on the trait of interest. In the present study, GEI patterns for the selected quality and mixograph traits were studied using the Additive Main Effects and Multiplicative Interaction (AMMI) model. Two biparental wheat populations consisting of 145 and 175 RILs were evaluated in six environments. The environment was the dominant source of variation for grain protein content (GPC), wet gluten content (WGC), and test weight (TW), accounting for approximately 40% to 85% of the total variation. The pattern was less consistent for mixograph traits for which the dominant source of variation has been shown to be trait and population-dependent. Overall, GEI has been shown to play a more important role for mixograph traits compared to other quality traits. Inspection of the AMMI2 biplot revealed some broadly adapted RILs, among which, MG124 is the most interesting, being the prevalent "winner" for GPC and WGC, but also the "winner" for non-correlated trait TW in environment SB10.

**Keywords:** wheat quality; mixograph; biparental population; GEI; AMMI; EM-AMMI

## 1. Introduction

Before release and widespread use for human consumption, wheat cultivars must possess suitable end-use quality. However, baking quality improvement is one of the most demanding objectives in wheat breeding, since the majority of quality traits exhibit complex inheritance patterns. Wheat baking quality could be evaluated by a large number of traits, which are generally controlled by many minor-effect and a few large-effect genes [1]. One of the most important factors affecting dough characteristics is gluten content and its strength [2,3]. Gluten is the most abundant wheat protein, which contributes to approximately 75–80% of total grain protein content (GPC) [4]. Such a large abundance consequently causes a high positive correlation of GPC and gluten content with other quality traits [5]. Structurally, gluten is a complex network of monomeric gliadins and polymeric subunits of glutenin [6,7], the quality and strength of which is determined by the proportion of its components and their quality [8,9], rather than by the total GPC or gluten content. Among all gluten components, high molecular weight glutenin subunits (HMW-GS) have the greatest impact on bread quality [8]. GPC is usually used as an indicator of baking quality [10], while wet gluten content (WGC) illustrates dough water absorption and its ability to form a gluten network, hence indicating the stability of the dough [11]. Among

other grain quality traits, test weight (TW) is often used as a predictor of flour extraction and thus wheat quality [12]. Taking into account often costly and time-consuming phenotyping for wheat baking quality, as well as the existence of complex interaction between wheat proteins and other components, such as pentosans and puroindolins, predictability of dough baking quality may be very difficult [13,14]. Therefore, in addition to the qualitative and quantitative composition analyses, rheological tests are required in order to assess baking quality more accurately. Rheological tests simulate the baking performance of the dough, evaluating its mixing and viscoelastic properties. Different devices can be used to perform rheological tests, i.e., farinograph, extensograph, and mixograph. Mixograph is a dough mixer that creates dough rheological profile based on numerous variables, which together with grain protein and gluten content, can give a reliable estimation of baking quality [15]. It is used to obtain general information about dough mixing and behavior during development, as well as the strength of the dough [16,17]. Due to the small amount of flour required (2, 10, or 35 g) and relatively fast interpretation of the obtained results, mixograph is highly suitable for use in plant breeding especially in early generations [18].

The success of wheat quality improvement depends on the ability to develop a genotype with both superior performance and high stability of the quality traits. In this context, one of the major challenges in plant breeding is the existence of genotype-by-environment interaction (GEI). In order to achieve more effective selection in wheat breeding programs, it is of the utmost importance to understand how genotype, environment, and GEI each contribute to the trait of interest [19]. Differential response of genotypes depending on the environmental conditions has already been determined for the majority of the wheat grain quality traits [20–23]. The greatest part of the phenotypic variation for GPC is due to non-genetic factors among which a strong influence of environment has been well documented [20,24,25]. Although gluten content shows a positive correlation with GPC, it is considered that for its qualitative characteristics genotype plays the most important role [26]. Compared to GPC, dough rheological traits are showed to be less influenced by GEI in some cases [27–29], while other research showed that variation due to GEI is equal or higher than variation caused by genotype or environment alone [20,23].

One of the most widely used methods for GEI analysis is the Additive Main Effects and Multiplicative Interaction (AMMI) model. It represents the combination of analysis of variance (ANOVA) procedure for main effects and the singular value decomposition (SVD) of GEI term [30]. Although most commonly used in multi-environment analyses of the most important traits such as wheat yield, AMMI model was also used to analyze GEI and stability of wheat quality traits [20,31,32]. In order to investigate GEI, the phenotypic performance of a collection of genotypes must be assessed in multiple environments. In plant breeding, recombinant inbred lines (RILs) represent a collection of genotypes with a noteworthy source of genetic diversity. However, in order to select RILs with wide trait adaptation, breeders should be able to reveal GEI effects underlying the trait of interest and to identify stable genotypes. The ability of AMMI model to explore GEI and to identify stable high-yielding genotypes in wheat populations consisting of RILs has already been proven in several studies [33–35]. Rodriguez et al. [36] concluded that RILs were more stable and tolerable to variable environmental conditions when compared to landraces and varieties. Although still not numerous, studies have shown that AMMI model can be used to successfully analyze GEI and its structure for wheat quality traits in RIL populations [37]. Among the traits tested in RIL populations, GPC is shown to be under the largest influence of environment, as well as of joint effects of environment and GEI (>98% and >99% of contribution to the total sum of squares, respectively) [38]. The predominant influence of environment, with more than 90% of phenotypic variation due to the joint influence of environment and GEI was reported by Elangovan et al. [39] for both GPC and TW. Furthermore, Prashant et al. [40] examined sources of variation affecting mixograph traits and determined a similar pattern, namely, the substantial contribution of environment and GEI to phenotypic variation of mixograph traits as well. On the other hand, there is some evidence that the AMMI model is not applicable with the same efficiency to all quality

traits. Specifically, regardless of the high and significant effect of GEI, AMMI model was not particularly successful in identifying stable genotypes for loaf volume [41]. Nevertheless, AMMI model has been able to successfully identify specifically adapted as well as stable wheat genotypes for the majority of quality traits examined in RIL populations [38].

The objectives of this study were to investigate transgressive and GEI patterns for selected quality and mixograph traits in two biparental wheat populations derived from the crosses of pairs of parental cultivars that either did not differ at all or differed in all HMW-GS, which play a key role in breeding for improved bread-making quality.

## 2. Materials and Methods

### 2.1. Plant Material

Two biparental (RIL) winter wheat populations were used in this study. The BK and MG population were derived from the Bezostaya-1 × Klara and Monika × Golubica crosses, respectively. In BK combination parental cultivars differed in all HMW-GS, while in MG combination parental cultivars did not differ in any HMW-GS [42]. The BK combination represented an example of crossing that is often applied in practical breeding programs where parents of significant phenotypic divergence are crossed, while the MG combination was chosen to narrow phenotypic variation to genetic factors excluding the differences in HMW-GS. After crossing and selfing, plants were randomly selected up to the F7 generation. The BK and MG populations consisted of 145 and 175 genotypes, respectively, including parental cultivars. Klara, Monika, and Golubica are high-yielding cultivars of good bread-making quality developed at the Agricultural Institute Osijek, while Bezostaya-1 is a Russian cultivar with good technological grain quality.

### 2.2. Description of Field Trials

The field trials were conducted during three consecutive years (2009–2011) at two locations in Croatia—Osijek (OS) and Slavonski Brod (SB), and both populations were evaluated in these six environments (location–year combinations). Soil type represented at locations Osijek and Slavonski Brod is eutric cambisol and eugley, respectively. Osijek and Slavonski Brod locations are classified as having a moderately warm and rainy oceanic climate (Cfb) by the Köppen–Geiger climate classification. A summary of meteorological data for both locations during three growing seasons is presented in Tables S1–S3 (Supplementary Materials). On average, no substantial difference was observed in daily temperatures and soil temperatures at 5 cm soil depth between Osijek and Slavonski Brod. The biggest difference in total rainfall between locations was observed during the 2008/2009 season, while no substantial difference was observed during two remaining seasons. In overall, the highest amount of precipitation was recorded during the 2009/2010 season.

In each environment, the field trial was set as a row–column design with two replicates, in 16 rows by 19 or 22 columns (for BK or MG population, respectively). The initial plot size was 4.86 m$^2$, and prior to harvesting, the front and back end of plots were trimmed to final net plot size of 2.7 m$^2$ for all genotypes and in all trials. Basic fertilization prior to planting was applied by adding 100 kg ha$^{-1}$ of urea (46% N) and 300 kg ha$^{-1}$ NPK (7:20:30) at both locations and all seasons. The N applied at top-dressings was 27 kg N ha$^{-1}$ at tillering and stem extension growth stages, respectively. Total amount of applied macronutrients was 121 kg N ha$^{-1}$, 60 kg P$_2$O$_5$ ha$^{-1}$ and 90 kg K$_2$O ha$^{-1}$. All other cultural practices including application of herbicides, insecticides, and fungicides to control major weeds, insects and foliar diseases were typical for commercial wheat production in Croatia. In the trials with BK population, parental cultivars were sown twice per replicate and another control (L84-2004) was added to both trial populations, to fill the grid. Due to the low flour sample quality and possible unreliability of obtained mixograph results, data collected for the BK population at the location Slavonski Brod in the year 2011 were not used in the analysis. In the text, tables, and graphical representations that follow, environments will be denoted using the location-year combination abbreviations: OS09 (Osijek—2009),

OS10 (Osijek—2010), OS11 (Osijek—2011), SB09 (Slavonski Brod—2009), SB10 (Slavonski Brod—2010), and SB11 (Slavonski Brod—2011).

### 2.3. Phenotyping

The TW and GPC traits were determined by near-infrared spectroscopy on whole grains using the Infratec 1221 Grain Analyzer and reported at 14% moisture basis, and expressed in kg hL$^{-1}$ and %, respectively. Prior to WGC and mixograph analyses, grain samples were tempered and milled using a laboratory mill. The WGC was determined according to ICC standard method No 155 using the Glutomatic 2200 Gluten System and Glutomatic Centrifuge 2015, Perten, using a 10 g flour sample. The WGC is expressed as a percentage of mass relative to the initial sample mass, and calculated according to Equation (1):

$$\text{WGC (\%)} = \frac{\text{total wet gluten (g)}}{10\ \text{g}} \times 100 \tag{1}$$

Dough rheology was assessed using the Swanson and Working Mixograph (National MFG Co., National Manufacturing Company, Lincoln, NE, USA). The amount of flour required for analysis is determined according to the sample protein amount and sample moisture. Flour moisture was measured using a Mettler Toledo HR83 moisture analyzer. Prior to mixing, the required amount of water was added to the sample, calculated according to the Equation (2):

$$\% \text{ abs} = (1.5 \times \% \text{ protein} + 43.6) \times 10\ [\text{mL}] \tag{2}$$

The results of the analysis were processed using MixSmart software (v 3.40). The best repeatability was determined for the variables of the central curve, which provide a comprehensive view of the optimal dough development. Therefore, the following variables were used as input for statistical analysis: MPT (midline peak time [min])—time required to achieve maximum dough resistance, i.e., time required for optimal dough development; MTW (midline curve tail width [%])—width of the peak at the end of the mixing period that indicates the consistency and stability of the dough at the end of mixing process; MTI (midline curve tail integral)—area below the midline curve from the starting point to the end of the mixing process that describes energy used during the mixing process; and MPH (midline peak height [%])—indicates the dough strength [40].

### 2.4. Statistical Analysis

At the first stage of the analysis, pooled data from individual trials were analyzed using the mixed model:

$$\text{Y} = \text{G} + \text{E} + \text{G·E} + \text{REP·E} \ : \ \text{ROW·REP·E} + \text{COL·REP·E} \tag{3}$$

that included the fixed effects of genotype (G), environment (G), GEI (G·E), and replicates within environments (REP·E), as well as the random effects of rows and columns within replicates within environments (ROW·REP·E and COL·REP·E, respectively). By allowing for all nested effects to vary across environments and removing the zero-variance effects, the optimal model was built using the Wald test and AIC (Akaike's information criterion) as selection criteria. Predicted values for all genotype-environment combinations were then taken as the input for the second stage of the analysis. The first stage of the analysis was performed within R environment [43], using the commercial package "asreml" [44] and freely available companion package "asremlPlus" [45].

The second stage of analysis began with an assessment of Pearson's correlations between the traits within and across available environments, using estimated values for all genotype-environment combinations from the first stage of the analysis. Mean values for parental cultivars together with mean values and ranges for the RILs were calculated

for all traits within as well as across environments. Matrix of genotype by environment estimates was decomposed using the AMMI model:

$$Y = G + E + (G \cdot E)_{pattern} + (G \cdot E)_{noise} \tag{4}$$

where Y is the predicted value of the dependent variable, G is the genotypic effect, E is the environmental effect, and interaction effect G·E is divided into selected AMMI model estimate—$(G \cdot E)_{pattern}$ and discarded residual—$(G \cdot E)_{noise}$. $(G \cdot E)_{pattern}$ effect is the sum of the appropriate (for a certain genotype-environment combination) matrix elements for k selected AMMI axes, where each matrix element is the product of the singular value for axis k ($\lambda_k$) and the appropriate elements of genotypic and environmental vectors for the same axis ($\gamma_k$ and $\delta_k$, respectively) [46,47].

In order to handle missing data that occurred for mixograph traits in both populations, the analysis for those traits was performed by using the Expectation–Maximization AMMI (EM–AMMI) algorithm according to Paderewski [48]. Three different approaches were used for the selection of terms that should be retained in the final model: (1) Simple parametric bootstrap (SPB) [49], (2) $F_R$-test [50], and (3) Leave-one-out cross-validation (LOO CV) procedure [48]. The contributions of genotype, environment, and GEI terms to the total variance were calculated as the ratio of the sum of squares (SS) of the corresponding term and the total SS, and expressed as a percentage [51]. The contributions of interaction principal component axis (IPCA) scores to GEI were calculated on the same principle, using the corresponding IPCA SS and GEI SS.

Instead of standard AMMI biplots, AMMI1 (main effects vs. IPCA1), and AMMI2 (IPCA1 vs. IPCA2), AMMI dissection of GEI patterns was visualized rather by using a modified version of AMMI2 biplot. It represents an attempt to combine properties of both standard types by adding the main effects to the AMMI2 biplot using a color scale. In order to avoid point cluttering, genotypic and environmental IPCA scores were calculated by applying different scaling (by multiplying their appropriate eigenvectors with two times square root and half square root of eigenvalues, respectively).

Second stage of the analysis was likewise performed within R environment [43], using the following packages: "corrplot" [52], "Hmisc" [53], "reshape2" [54], "dplyr" [55], "ggplot2" [56], and "ggrepel" [57]. An R function "EM.AMMI" [48] has been applied for the data imputation prior to AMMI analysis using two IPCAs, 1000 iterations, and precision level of 0.001 to run the EM–AMMI procedure. For the SPB test the adjusted R script from Forkman and Piepho [49] was used and the probability value of the test was obtained using 100,000 bootstrap samples. The $F_R$-test was performed as suggested by Piepho [50] using the degrees of freedom estimated for IPCA terms according to Gollob [58]. An R function "CV.LOO" [48] was used to perform LOO CV procedure using four (BK dataset) and five (MG dataset) IPCAs, and the permissible minimum number of observed values (MNO) in each row and each column of the data matrix set to three.

## 3. Results

### 3.1. The First Stage of the Analysis

Starting with the full model including separate nested effects for each environment, for each trait and population, models were gradually reduced until the optimal model was reached. The nested effects structure in selected optimal models is shown in the Supplementary Table S4. For random effects full model was reduced by removing all zero-variance effects (if any); there was one case where one of the effects was reduced to a single effect for all environments and the other completely removed from the model, thus providing the substantial reduction of AIC (MTW in MG). The fixed effect of replicates was truncated by keeping it in the model only for environments in which the Wald test was significant if that resulted in lower AIC compared to the model with full effect. Overall, no obvious pattern could be observed, as the optimal models tend to be specific for each trait/population combination.

*3.2. Transgressive Segregation in Quality Traits*

Overall descriptive statistics for seven traits of both populations are shown in Table 1, while Tables S5 and S6 (Supplementary Materials) are showing descriptive statistics per environment for BK and MG population, respectively. Means for parental cultivars are followed by means and ranges of RILs. Transgressive segregants are denoted as "positive" if they exhibited values higher than the parental cultivar with higher trait value, and as "negative" if their values were lower compared to the parental cultivar with lower trait value. Across environments, Bezostaya-1 had a higher mean compared to Klara for all traits except TW, MTI, and MPH, even though the difference in means for these two parental cultivars was not highly pronounced. The range of BK RILs was much wider than the range of parental cultivars, and both positive and negative transgressive segregants are found for all traits examined (in the range of 38.46–77.62% and 5.59–46.85% for positive and negative segregants, respectively). This may suggest the presence of increaser as well as decreaser alleles for quality traits in both parents. Mean values for BK RILs were similar to parental means for all traits, except for the mixograph traits MPT and MTW, the mean values of which were higher than the higher-performing parental cultivar (Table 1). These were also the traits for which the highest proportion of positive and the lowest proportion of negative segregants was observed, while for all the other traits across environments positive and negative segregants were equally represented. In the MG population, Golubica can be identified as the parent with a higher value for most of the traits, both within and across environments (Table 1). Mean values for MG RILs were within ranges of parental values for all traits as they were much wider than ranges between Bezostaya-1 and Klara. Consequently, a lower ratio of both positive and negative transgressive segregants was noticed in the MG population compared to the BK population in general. Trait-wise, the lowest ratios for both segregant groups were observed for mixograph traits MTI and MPH, while the largest disproportion between them was recorded in WGC where there were 5.4 times more positive than negative transgressive segregants. Considering individual environments, it is noticeable that means for MG RILs vary substantially between environments (Table S3 in Supplementary Materials). Interestingly, the strongest predominance of negative segregants over positive ones is recorded for mixograph traits MTI and MPH in the OS11 environment, where more than 90% negative and approximately 1% positive segregants were present. On the contrary, the same two traits showed the highest predominance of positive over negative segregants in environment SB11, with approximately 50% of positive and almost no negative segregants present.

**Table 1.** Summary of parental means, RIL means, and ranges, and rates of transgressive segregants across environments for seven quality traits assessed in BK and MG RIL wheat populations.

| Trait | Parental Cultivars Mean | | RILs | | | Positive Transgressive Segregants [3] | | Negative Transgressive Segregants [4] | |
|---|---|---|---|---|---|---|---|---|---|
| | P1 [1] | P2 [2] | Min | Mean | Max | N | % | N | % |
| BK population | | | | | | | | | |
| GPC [5] | 14.3 | 13.9 | 10.6 | 14.0 | 17.5 | 55 | 38.5 | 60 | 42.0 |
| WGC | 34.7 | 34.1 | 20.5 | 34.1 | 43.9 | 61 | 42.7 | 62 | 43.4 |
| TW | 79.1 | 79.8 | 64.8 | 79.5 | 86.9 | 71 | 49.7 | 46 | 32.2 |
| MPT | 5.2 | 4.6 | 1.3 | 5.4 | 10.0 | 86 | 60.1 | 19 | 13.3 |
| MTW | 20.1 | 18.8 | 10.8 | 21.3 | 35.1 | 111 | 77.6 | 8 | 5.6 |
| MTI | 359.6 | 367.2 | 237.3 | 361.3 | 504.9 | 65 | 45.5 | 66 | 46.2 |
| MPH | 41.6 | 42.4 | 25.5 | 41.8 | 58.9 | 66 | 46.2 | 67 | 46.9 |
| MG population | | | | | | | | | |
| GPC | 13.0 | 14.1 | 11.0 | 13.8 | 17.3 | 56 | 32.4 | 21 | 12.1 |
| WGC | 29.9 | 35.3 | 21.8 | 33.5 | 43.9 | 27 | 15.6 | 5 | 2.9 |
| TW | 79.3 | 80.4 | 65.7 | 79.6 | 86.2 | 46 | 26.6 | 70 | 40.5 |
| MPT | 4.6 | 4.9 | 1.6 | 4.8 | 9.1 | 95 | 54.9 | 63 | 36.4 |

**Table 1.** *Cont.*

| Trait | Parental Cultivars Mean | | RILs | | | Positive Transgressive Segregants [3] | | Negative Transgressive Segregants [4] | |
|---|---|---|---|---|---|---|---|---|---|
| | P1 [1] | P2 [2] | Min | Mean | Max | N | % | N | % |
| MTW | 9.3 | 17.9 | 4.4 | 15.2 | 45.9 | 46 | 26.6 | 19 | 11.0 |
| MTI | 347.5 | 438.8 | 285.5 | 382.8 | 527.8 | 3 | 1.7 | 10 | 5.8 |
| MPH | 41.5 | 52.2 | 30.8 | 45.4 | 73.2 | 5 | 2.9 | 18 | 10.4 |

[1] Bezostaya-1 and Monika for BK and MG population, respectively. [2] Klara and Golubica for BK and MG population, respectively. [3] RILs that exhibited values higher than the parental cultivar with higher trait value. [4] RILs that exhibited values lower than the parental cultivar with lower trait value. [5] Abbreviations: grain protein content (GPC), wet gluten content (WGC), test weight (TW), midline peak time (MPT), midline curve tail width (MTW), midline curve tail integral (MTI), midline peak height (MPH).

### 3.3. Phenotypic Correlations between Quality Traits

A summary of Pearson's correlation coefficients for both populations is presented in a form of correlograms across environments (Figure 1) and within environments (Figures S1 and S2 in Supplementary Materials). Very high positive correlations were observed across environments between GPC and WGC ($r \geq 0.89$), and between MTI and MPH ($r \geq 0.91$), and they were constant in both populations as well as within environments. Although not being very strong, negative correlations were observed in both populations between GPC/WGC and TW across environments. On the other hand, within environments, these correlations were mostly positive for the BK population with the lowest values recorded in the SB10 environment, while for the MG population almost no correlation was observed between these traits, except in the SB11 environment where correlations were positive but very weak ($r \sim 0.2$). Similarly, TW exhibited negative but low correlations with all four mixograph traits in the BK population, while in the MG population these correlations were mostly positive across environments. Substantial variations in terms of correlation strength and direction were observed for these trait pairs in both populations within environments. GPC and WGC showed positive correlations with mixograph traits MTI and MPH across environments in both populations, with the difference that correlations were much higher in BK compared to the MG population. These correlation patterns were also consistent within environments but showed higher variation in the case of the MG population (ranging between 0.18 and 0.66). In contrast, the correlations of the other two mixograph traits (MPT and MTW) with GPC/WGC exhibited quite different trends. MPT was weakly correlated with both GPC and WGC with opposite signs in two populations; MTW was moderately negatively correlated with them in the MG population, but not correlated in the BK population. This pattern was highly inconsistent within environments. Variations between environments observed for some trait pairs (mostly between TW and other traits), especially regarding the direction of correlations, should indicate a strong environmental impact.

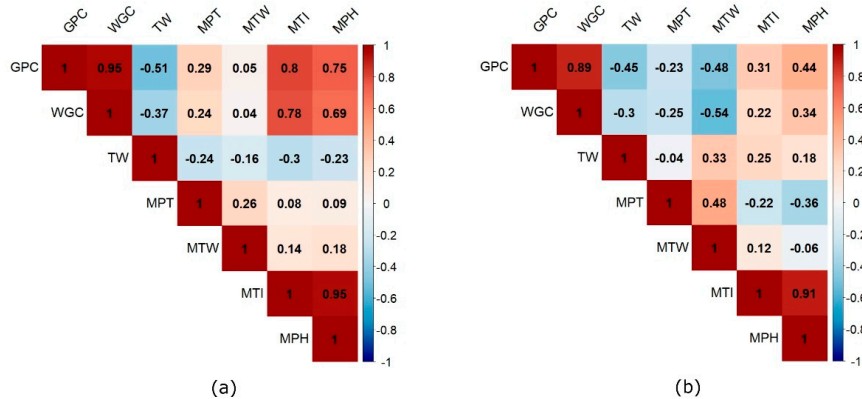

(a)             (b)

**Figure 1.** Pearson's correlation coefficients across environments for (**a**) BK population and (**b**) MG population.

### 3.4. Selection of the Appropriate AMMI Model

In the model selection process, i.e., deciding upon how many significant IPCA terms should be retained in the model, three different methods were compared: the SPB method, $F_R$-test, and LOO CV method. For all four mixograph traits in both populations, a small proportion of missing data occurred (<1%), so the presence of missing data implied the use of the EM–AMMI algorithm. Unlike LOO CV, which can be embedded into EM–AMMI analysis, for two other model selection methods, this creates the drawback caused by preselecting the model, i.e., beforehand decision upon the number of axes to be retained in the model. To resolve this issue, tests were first applied on the EM–AMMI imputed data based on all possible models, from AMMI0 to AMMIF (although for AMMIF algorithm never converged). Since no difference was observed regards the model selected using the simple parametric bootstrap method and $F_R$-test (Tables S7 and S8 in Supplementary Materials), all AMMI1 imputed data were used in all further calculations.

Tables 2 and 3 summarize test outcomes for all IPCA terms used, for the BK (4) and MG (5) population, stating their SS and contributions to GEI SS, as well as the test statistics and corresponding probability values for all methods applied. The optimal number of IPCA terms for SPB and $F_R$-test was determined by *p*-value taking into account all terms until a non-significant value is obtained (at the level of significance $p < 0.05$). The selection by the LOO CV method is based on Root Mean Square Prediction Difference (RMSPD), and the optimal model was the one with the lowest RMSPD. Generally, there is a large disagreement between the test outcomes. In exactly half of the trait-population combinations, SPB and $F_R$-test selected the same AMMI model, while in the other half they selected the models that could substantially differ in the number of axes retained. $F_R$-test was overall more liberal, and more prone to the tendency to declare all tests significant. On the other side, there is the LOO CV, as the most conservative criterion. It selected the additive model (AMMI0) as the most appropriate one for all except one trait in each population (MTI in BK, and MTW in MG population).

**Table 2.** Results of tests for IPCA terms for seven quality traits assessed in BK population.

| IPCA | Sum of Squares | | | Simple Bootstrap | | $F_R$-Test | | Cross-Validation |
|---|---|---|---|---|---|---|---|---|
| | IPCA SS | % | Total % | *T* | *p* Value | *F* | *p* Value | *RMSPD* |
| | | | | GPC | | | | |
| 0 | | | | | | | | 14.7 |
| 1 | 58.0 | 42.6 | 42.6 | 0.43 | 0.000 | 2.16 | 0.000 | 21.1 |
| 2 | 40.0 | 29.4 | 71.9 | 0.51 | 0.000 | 2.05 | 0.000 | 49.9 |
| 3 | 21.6 | 15.8 | 87.8 | 0.57 | 0.315 | 1.27 | 0.076 | 52.0 |
| 4 | 16.7 | 12.3 | 100.0 | | | | | |
| | | | | WGC | | | | |
| 0 | | | | | | | | 44.4 |
| 1 | 524.7 | 42.2 | 42.2 | 0.42 | 0.000 | 2.13 | 0.000 | 54.1 |
| 2 | 343.1 | 27.6 | 69.8 | 0.48 | 0.001 | 1.79 | 0.000 | 105.9 |
| 3 | 192.8 | 15.5 | 85.3 | 0.51 | 0.955 | 1.04 | 0.413 | 197.9 |
| 4 | 183.2 | 14.7 | 100.0 | | | | | |
| | | | | TW | | | | |
| 0 | | | | | | | | 39.6 |
| 1 | 562.4 | 56.9 | 56.9 | 0.57 | 0.000 | 3.85 | 0.000 | 94.1 |
| 2 | 284.1 | 28.7 | 85.6 | 0.67 | 0.000 | 3.91 | 0.000 | 97.7 |
| 3 | 89.9 | 9.1 | 94.7 | 0.63 | 0.007 | 1.69 | 0.001 | 118.0 |
| 4 | 52.5 | 5.3 | 100.0 | | | | | |
| | | | | MPT | | | | |
| 0 | | | | | | | | 39.3 |
| 1 | 400.2 | 41.2 | 41.2 | 0.41 | 0.000 | 2.05 | 0.000 | 46.9 |
| 2 | 268.8 | 27.7 | 68.8 | 0.47 | 0.001 | 1.73 | 0.000 | 102.3 |
| 3 | 209.9 | 21.6 | 90.4 | 0.69 | 0.000 | 2.23 | 0.000 | 108.8 |
| 4 | 93.3 | 9.6 | 100.0 | | | | | |

**Table 2.** *Cont.*

| IPCA | Sum of Squares | | | Simple Bootstrap | | $F_R$-Test | | Cross-Validation |
|---|---|---|---|---|---|---|---|---|
| | IPCA SS | % | Total % | *T* | *p* Value | *F* | *p* Value | *RMSPD* |
| | | | | MTW | | | | |
| 0 | | | | | | | | 74.1 |
| 1 | 1774.3 | 48.7 | 48.7 | 0.48 | 0.000 | 2.68 | 0.000 | 102.0 |
| 2 | 800.3 | 22.0 | 70.7 | 0.43 | 0.061 | 1.46 | 0.003 | 119.9 |
| 3 | 571.6 | 15.7 | 86.4 | 0.54 | 0.684 | 1.14 | 0.215 | 235.5 |
| 4 | 496.8 | 13.6 | 100.0 | | | | | |
| | | | | MTI | | | | |
| 0 | | | | | | | | 796.8 |
| 1 | 126404.6 | 31.3 | 31.3 | 0.32 | 0.255 | 1.37 | 0.009 | 1043.1 |
| 2 | 116513.4 | 28.8 | 60.1 | 0.41 | 0.210 | 1.35 | 0.016 | 1983.6 |
| 3 | 94460.8 | 23.4 | 83.5 | 0.59 | 0.120 | 1.40 | 0.024 | 133.1 |
| 4 | 66703.0 | 16.5 | 100.0 | | | | | |
| | | | | MPH | | | | |
| 0 | | | | | | | | 102.6 |
| 1 | 1999.9 | 30.2 | 30.2 | 0.30 | 0.606 | 1.27 | 0.036 | 128.3 |
| 2 | 1866.4 | 28.1 | 58.3 | 0.40 | 0.296 | 1.32 | 0.026 | 157.6 |
| 3 | 1466.6 | 22.1 | 80.4 | 0.53 | 0.792 | 1.11 | 0.275 | 252.0 |
| 4 | 1300.9 | 19.6 | 100.0 | | | | | |

**Table 3.** Results of tests for IPCA terms for seven quality traits assessed in MG population.

| IPCA | Sum of Squares | | | Simple Bootstrap | | $F_R$-Test | | Cross-Validation |
|---|---|---|---|---|---|---|---|---|
| | IPCA SS | % | Total % | *T* | *p* Value | *F* | *p* Value | *RMSPD* |
| | | | | GPC | | | | |
| 0 | | | | | | | | 20.7 |
| 1 | 108.2 | 36.6 | 36.6 | 0.37 | 0.000 | 2.25 | 0.000 | 42.2 |
| 2 | 71.9 | 24.4 | 61.0 | 0.38 | 0.000 | 1.83 | 0.000 | 69.2 |
| 3 | 45.1 | 15.3 | 76.3 | 0.39 | 0.376 | 1.26 | 0.035 | 84.0 |
| 4 | 40.5 | 13.7 | 90.0 | 0.58 | 0.124 | 1.35 | 0.024 | 75.1 |
| 5 | 29.6 | 10.0 | 100.0 | | | | | |
| | | | | WGC | | | | |
| 0 | | | | | | | | 62.8 |
| 1 | 855.4 | 31.6 | 31.5 | 0.32 | 0.000 | 1.79 | 0.000 | 69.3 |
| 2 | 748.9 | 27.6 | 59.2 | 0.40 | 0.000 | 1.99 | 0.000 | 178.6 |
| 3 | 432.3 | 15.9 | 75.1 | 0.39 | 0.385 | 1.26 | 0.036 | 295.1 |
| 4 | 405.4 | 15.0 | 90.1 | 0.60 | 0.028 | 1.49 | 0.005 | 262.5 |
| 5 | 268.4 | 9.9 | 100.0 | | | | | |
| | | | | TW | | | | |
| 0 | | | | | | | | 44.7 |
| 1 | 957.0 | 69.7 | 69.7 | 0.70 | 0.000 | 8.92 | 0.000 | 48.6 |
| 2 | 151.6 | 11.0 | 80.7 | 0.36 | 0.001 | 1.68 | 0.000 | 69.2 |
| 3 | 116.9 | 8.5 | 89.2 | 0.44 | 0.009 | 1.55 | 0.000 | 90.3 |
| 4 | 77.3 | 5.6 | 94.8 | 0.52 | 0.864 | 1.07 | 0.319 | 115.7 |
| 5 | 71.1 | 5.2 | 100.0 | | | | | |
| | | | | MPT | | | | |
| 0 | | | | | | | | 35.6 |
| 1 | 352.0 | 39.8 | 39.8 | 0.40 | 0.000 | 1.93 | 0.000 | 50.4 |
| 2 | 154.0 | 17.4 | 57.2 | 0.29 | 0.817 | 0.80 | 0.953 | 162.4 |
| 3 | 149.7 | 16.9 | 74.1 | 0.40 | 0.302 | 0.65 | 0.998 | 185.4 |
| 4 | 128.2 | 14.5 | 88.6 | 0.56 | 0.283 | 2.54 | 0.000 | 175.8 |
| 5 | 100.6 | 11.4 | 100.0 | | | | | |

**Table 3.** *Cont.*

| IPCA | Sum of Squares | | | Simple Bootstrap | | $F_R$-Test | | Cross-Validation |
|---|---|---|---|---|---|---|---|---|
| | IPCA SS | % | Total % | *T* | *p* Value | *F* | *p* Value | *RMSPD* |
| | | | | MTW | | | | |
| 0 | | | | | | | | 114.4 |
| 1 | 3996.4 | 43.2 | 43.2 | 0.43 | 0.000 | 2.24 | 0.000 | 114.3 |
| 2 | 1726.7 | 18.7 | 61.9 | 0.31 | 0.402 | 0.87 | 0.858 | 385.1 |
| 3 | 1391.4 | 15.0 | 76.9 | 0.39 | 0.332 | 0.64 | 0.998 | 336.6 |
| 4 | 1165.1 | 12.6 | 89.5 | 0.54 | 0.563 | 2.34 | 0.000 | 454.4 |
| 5 | 973.6 | 10.5 | 100.0 | | | | | |
| | | | | MTI | | | | |
| 0 | | | | | | | | 1078.0 |
| 1 | 255111.6 | 32.0 | 32.0 | 0.32 | 0.000 | 1.38 | 0.003 | 1182.6 |
| 2 | 195556.4 | 24.5 | 56.5 | 0.36 | 0.001 | 1.11 | 0.211 | 1473.6 |
| 3 | 137022.0 | 17.2 | 73.7 | 0.40 | 0.311 | 0.65 | 0.998 | 2656.3 |
| 4 | 114022.2 | 14.3 | 88.0 | 0.54 | 0.509 | 2.38 | 0.000 | 2750.6 |
| 5 | 95698.0 | 12.0 | 100.0 | | | | | |
| | | | | MPH | | | | |
| 0 | | | | | | | | 150.9 |
| 1 | 4709.2 | 30.1 | 30.1 | 0.30 | 0.001 | 1.26 | 0.025 | 172.8 |
| 2 | 3671.7 | 23.5 | 53.6 | 0.34 | 0.027 | 1.00 | 0.506 | 217.1 |
| 3 | 2779.2 | 17.8 | 71.4 | 0.38 | 0.535 | 0.61 | 0.999 | 494.6 |
| 4 | 2562.1 | 16.4 | 87.8 | 0.57 | 0.155 | 2.68 | 0.000 | 684.3 |
| 5 | 1903.4 | 12.2 | 100.0 | | | | | |

### 3.5. GEI Patterns in Quality Traits

A preliminary insight to the biplot disclosure of GEI patterns is given in Table 4, showing relative contributions of the different AMMI2 model terms to the total variability. Besides population or trait-specific, some common general patterns can be observed as well. For three non-mixograph traits (GPC, WGC, and TW) the environment (E) was the dominant source of variation, consistently accounting for approximately three-quarters of the total variation in the BK population and varying between 40% and 85% in the MG population. The relative contributions of the two remaining effects are equally uniform in the BK population, where genotypic (G) variation is approximately twice as large as interaction (GEI) variation. This difference is reduced in the MG population, where G is approximately 1.5 times larger (GPC and WGC), or even smaller than GEI (TW). Finally, E is always much larger than the other two terms, except for GPC in the MG population where it is just slightly larger than G. Four mixograph traits could be roughly divided into two pairs that exhibit similar patterns. Dominant effect on MPT and MTW has GEI in BK and G in MG population. For this first pair of traits the E always has the weakest effect, except for MTW in the MG population, where the effect of E is slightly bigger than the effect of GEI. The other pair consists of MTI and MPH, characterized by the dominant effect of E in the BK and GEI in the MG population; the size of the contribution of G is always between the other two effects. The amount of GEI pattern captured by the first IPCA varies from 30% to 70%, which increases to a cumulative 54% to 86% for the first two IPCAs. The trait with the largest amount of pattern captured by either first IPCA or first and second together is TW in both populations.

**Table 4.** AMMI2 GEI patterns in both populations.

| Trait | BK Population | | | | | MG Population | | | | |
|---|---|---|---|---|---|---|---|---|---|---|
| | Contribution to Total SS (%) | | | Contribution to GEI (%) | | Contribution to Total SS (%) | | | Contribution to GEI (%) | |
| | G | E | GEI | IPCA1 | IPCA1 + IPCA2 | G | E | GEI | IPCA1 | IPCA1 + IPCA2 |
| GPC | 17.7 | 74.9 | 7.4 | 42.6 | 71.9 | 36.1 | 42.3 | 21.6 | 36.6 | 61.0 |
| WGC | 17.6 | 75.9 | 6.5 | 42.2 | 69.8 | 24.7 | 59.7 | 15.7 | 31.6 | 59.2 |
| TW | 18.3 | 71.6 | 10.2 | 56.9 | 85.6 | 6.9 | 84.9 | 8.1 | 69.7 | 80.7 |
| MPT | 26.8 | 11.0 | 62.2 | 41.2 | 68.8 | 47.1 | 12.6 | 40.4 | 39.8 | 57.2 |
| MTW | 39.8 | 9.2 | 50.9 | 48.7 | 70.7 | 45.5 | 33.7 | 20.8 | 43.2 | 61.9 |
| MTI | 29.9 | 45.9 | 24.2 | 31.3 | 60.1 | 31.2 | 23.7 | 45.2 | 32.0 | 56.5 |
| MPH | 30.7 | 40.3 | 29.0 | 30.2 | 58.3 | 36.7 | 15.7 | 47.5 | 30.1 | 53.6 |

Before inspecting the biplots, it is necessary to explain all the modifications made to the standard AMMI2 biplot. The color scale used on biplots indicates the mean value of the corresponding trait RILs, parental cultivars, and environments. RILs are designated by points; environments and parental cultivars by abbreviated labels (with or without a frame, respectively). "Winner" RILs are labeled and linked with their winning environments by a matching super/subscripted letter. The list of "winner" RILs within each environment is given in Supplementary Table S9. Interestingly, common "winner" RILs were not observed among traits that were not highly correlated in none of the two populations. In order to reduce the number of presented biplots, traits were paired based on correlations, the similarity of GEI patterns, and common "winners". One biplot from each pair is included in Figure 2 (BK population) or Figure 3 (MG population), and the other one moved to Figure S3 or Figure S4 in the Supplementary Materials. Generally, there are no clear patterns of distinction between neither locations nor years, so environmental variation is mostly due to the presence of some favorable or unfavorable combinations of locations and years. One such example is OS10, which is a favorable environment for the BK population for most of the traits (and unfavorable for TW); while the same statement is only partially applicable to the MG population. For breeding purposes, the focus is usually directed at finding generally or specifically adapted RILs. Proper candidates for selection as broadly adapted RILs are common "winners", such as BK012 for TW (Figure 2b), BK042 for MPT (Figure 2c), and MG124 for GPC and WGC (Figure S4a and Figure 3a, respectively). Especially interesting could be the last one, MG124, being the almost universal "winner" for GPC and dominant "winner" for its correlated trait WGC, but also "winner" for non-correlated trait TW in SB10. In some cases, the majority of "winning" RILs had low means of the corresponding trait, e.g., MTI in the BK population (Figure 2d) and MTW in the MG population (Figure 3c). Specifically adapted RILs are all exclusive "winners", and generally, all RILs that are located close to a certain environment. Out of all possible examples that could be found across all biplots, of particular interest could be those with closely matched color or completely mismatched color with the environment to which they are adapted. A mismatching example is high GPC line BK007 in the low GPC environment SB09, which due to high average protein content should be considered "broad" rather than "specifically" adapted RIL, capable to retain high protein content even in unfavorable environments. Before commenting on two matching cases for mixograph traits, it should be stressed that, for them, either low or high values are not immediately considered as desirable. Therefore, matching high values, i.e., MG016 with SB09 for MTW (Figure 3c) could be a negative feature if added interaction effect pushes already high values over the upper desirable limit. On the other side, matching low values, i.e., BK059 with SB09 for MTI, should generally be considered as positive features.

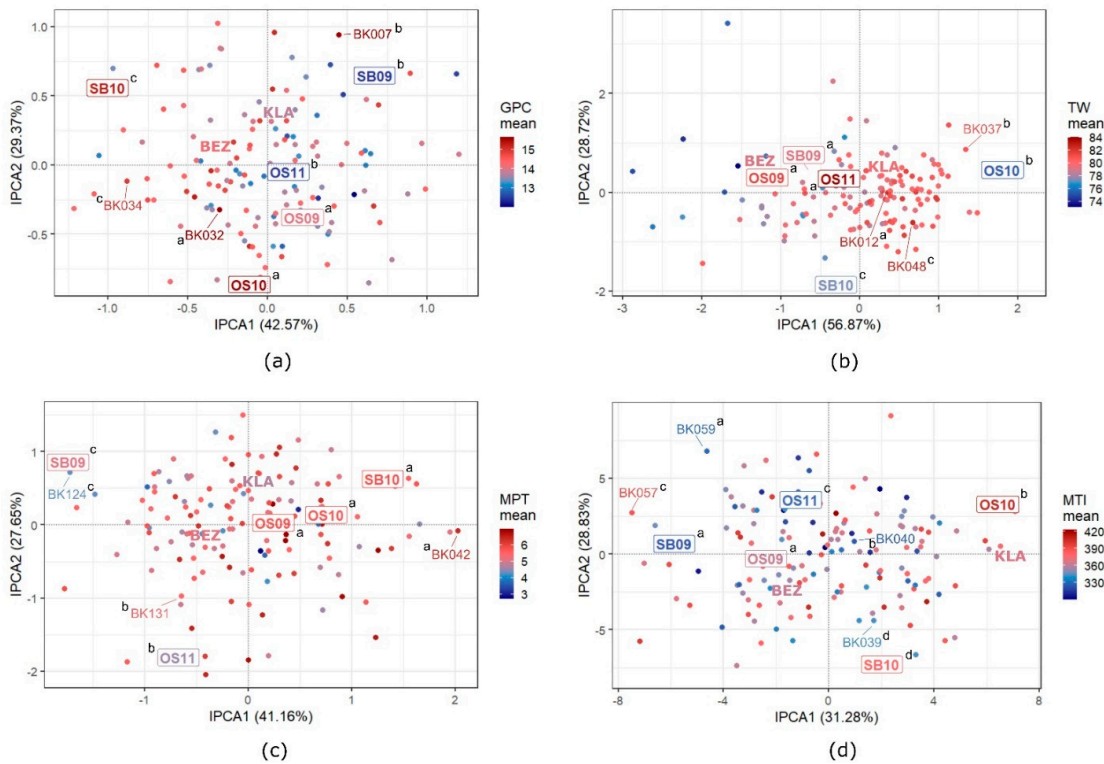

**Figure 2.** Modified AMMI2 biplots for (**a**) GPC, (**b**) TW, (**c**) MPT, and (**d**) MTI traits of BK RIL population. RILs are marked with dots, while environments and parental cultivars (Bezostaya-1 and Klara) are marked by abbreviated labels (with or without a frame, respectively). Within each environment one "winning" RIL is labeled and linked with its winning environment by a matching super/subscripted letter. The color indicates the mean value of the trait.

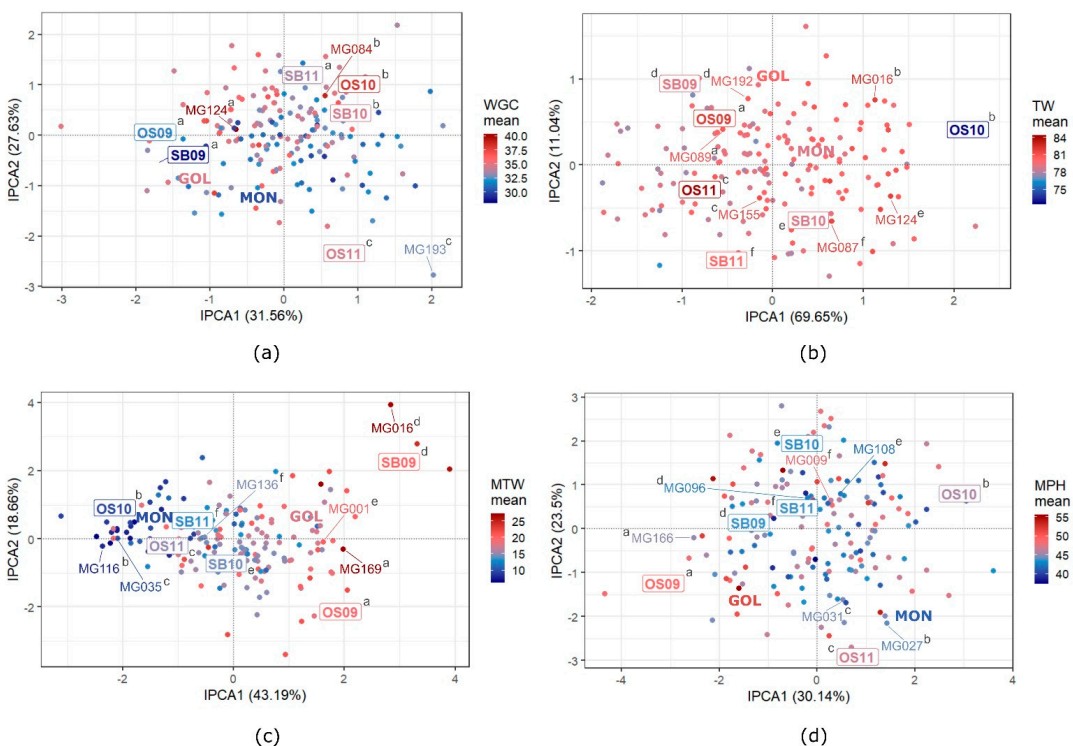

**Figure 3.** Modified AMMI2 biplots for (**a**) WGC, (**b**) TW, (**c**) MTW, and (**d**) MPH traits of MG RIL population. RILs are marked with dots, while environments and parental cultivars (Bezostaya-1 and Klara) are marked by abbreviated labels (with or without a frame, respectively). Within each environment, one "winning" RIL is labeled and linked with its winning environment by a matching super/subscripted letter. The color indicates the mean value of the trait.

## 4. Discussion

*4.1. Transgressive Segregation and Phenotypic Correlations among Quality Traits*

To produce high-quality dough and bread, wheat has to possess a suitable quality which is often assessed by protein and gluten content. Although the preferred value range of quality traits can vary considerably depending on the purpose of the wheat flour, for bread making GPC should be at least 12.5% [59]. WGC is an indicator of protein quality and according to Singh and Singh [60], its minimal value for wheat flour should be 24%. On the other hand, relevant research showed that gluten composition and its quality play a more important role compared to the quantity alone [7,8]. Horvat et al. [42] reported that, together with the composition of HMW-GS, the proportion of subunits must also be taken into account when assessing wheat quality. Research on the quality that included wheat cultivars explored in the present study showed that Bezostaya-1 was the cultivar of generally higher quality when compared to Monika and Golubica, which was attributed to the presence of 5 + 10 HMW-GS on D1 locus. However, regardless of possessing 2 + 12 HMW-GS type on D1 locus, which is generally associated with lower quality, Klara, Monika, and Golubica manifested very good quality. This elevated quality of cultivars with unfavorable HMW-GS was explained by higher proportion of total HMW-GS [42].

The extensibility and elasticity of gluten define its quality and consequently the quality of the dough, which can be estimated using the mixograph. Although it is difficult to exactly define desirable values for the mixograph traits as they vary depending on the type of population being studied and the purpose of the flour, generally it can be concluded that the good quality of dough is characterized by strong gluten which is indicated by longer optimal dough development time (MPT), greater consistency, and stability after mixing (MTW), higher energy used during mixing (MTI), and higher dough strength (MPH) [61]. The greatest advantage of using mixograph is that it requires a small amount of flour for analysis thus enabling the prediction of dough properties in early generation progenies.

In the present study, positive as well as negative transgressive segregants were detected in both populations although being generally more prevalent in the BK population. This result is expected considering that the parental cultivars Bezostaya-1 and Klara have different genetic bases of quality (different HMW-GS on all Glu-1 loci) [42]. For all examined traits, RIL means were positioned close to the parental means, except for the mixograph traits MPT and MTW within the BK population, the mean values of which were higher than the higher-performing parental cultivar Bezostaya-1. For these traits, this could be an indication of the presence of stronger epistatic effects. On the other hand, Monika and Golubica share the same genetic basis of quality, meaning that they possess the same type of HMW-GS on all Glu-1 loci; but show a higher phenotypic difference compared to Bezostaya-1 and Klara, resulting in a generally lower representation of segregants in MG RIL population. However, the occurrence of transgressive segregation for all quality traits examined in this study in both RIL populations suggests the presence of increaser and decreaser alleles in all four parental cultivars.

A high positive correlation between GPC and WGC recorded in both populations is expected, taking into account that gluten is the most abundant wheat protein and that the interrelatedness of these two traits is already well documented [4,5]. Additionally, in both populations, a high positive correlation was observed between mixograph traits MTI and MPH suggesting that higher energy must be used in the mixing process of the dough with higher strength and that this relationship is not dependent on the genetic background of wheat quality. When comparing correlations between WGC and mixograph traits for both populations in the present study, somewhat opposite patterns can be observed. The correlations between WGC and MTI/MPH were stronger in BK compared to the MG population suggesting that when at least one parental cultivar possesses HMW-GS associated with good bread-making quality, WGC can affect the dough strength. On the other hand, negligible positive and even negative correlations observed between WGC and MPT/MTW imply that optimal development time and the consistency of the dough do not

depend on the gluten quantity, but rather on its quality which is mostly influenced by the composition of HMW-GS [8]. These findings suggest that the parental genetic background of quality may not have an impact on traits such as GPC, WGC, and TW, if their phenotypic performance is already satisfactory, but may considerably affect mixograph traits, i.e., dough rheology. The importance of Glu-D1 in controlling mixograph traits has been confirmed in previous studies [61–63].

### 4.2. Selection of the Appropriate AMMI Model

The problem of finding the appropriate tests for IPCA terms is present in the literature on GEI over the last forty years. Numerous different tests were proposed, and some of them have already been compared in several published studies [49,50]. However, due to the specific nature of studies based on RILs (a large number of genotypes tested in just a few environments), it seemed interesting to employ three different tests in this study, and compare their outcomes. The LOO CV method turned out to be the most conservative, almost exclusively selecting the additive model (AMMI0), thus suggesting the complete absence of GEI. This does not seem to be a realistic conclusion, especially in cases where half or more of the total SS can be attributed to GEI. Out of the two remaining tests, $F_R$-test was slightly more liberal, tending to select more complex models, thus overfitting noise. This promotes the SPB test to the most appropriate one, which is in agreement with Forkman and Piepho's [49] conclusion. They have highly recommended the SPB test because it outperformed both $F_R$-test and cross-validation method in terms of performance power and probability of getting false-positive results, and accordingly, it was used in the present study as the criterion for the decision on the number of terms to be retained in the model. However, some caution should be taken when selecting more complex models (AMMI3 in this study), as they tend to overfit the noise [64], especially in the presence of missing data [65]. For the data sets with the larger amount of missing data, some more complex methods of data imputation [66] are probably more appropriate than the simple method used here [48].

### 4.3. GEI Patterns

According to the review by Williams et al. [27], the dominant effect of E for traits like GPC, WGC, and TW has been observed in numerous studies in the past, although they have cited more than a few exceptions to this general rule. However, rather than making comparisons over all sorts of GEI studies in wheat, it would make much more sense to search specifically for studies based on RILs, as they typically include a large number of genotypes tested over just a few environments. If some recent studies based on RILs are considered, there is almost no evidence for the dominant effect of E. It was reported for GPC only by Prashant et al. [40], while Echeverry-Solarte et al. [67] and Krishnappa et al. [38] detected equal effects of G and E for the same trait. On the contrary, in Goel et al. [68], the effect of E was smallest for GPC, WGC, and TW. The most common mixograph trait used in studies based on RILs is MPT, for which Prashant et al. [40] detected the dominant effect of E, the same as is in the BK population from the present study. In the other population, MG, the dominant effect for MPT is GEI, corresponding to the findings of Goel et al. [68]. The remaining effect, G, was found to be dominant for MPT in studies by both Echeverry-Solarte et al. [67] and Jin et al. [69]. Prashant et al. [40] have also detected dominant effects of GEI for MTW, and E for MTI, both corresponding to the same findings for the BK population in the present study. Generally, all the cited studies differ in many genotypic (type of parents, parental differences, etc.) as well as environmental (width of the range of environmental differences, climate conditions, etc.) factors, which all could be used as explanation why no common pattern could be established for any of the considered traits. However, GEI seems to be much more important for mixograph traits than for other quality traits (with one exception).

The use of standard AMMI biplots in studies involving a large number of genotypes can create an unreadable clutter of points or labels [34,70,71] unless a reduced set of lines

was selected for biplot representation [38]. While point cluttering can be avoided by different scaling of genotypic and environmental scores, label cluttering was prevented by labeling only parents and "winners". Once this problem was sorted out, the informativeness of standard biplots can be increased by carefully adding some extra features. Probably the most important modification is the addition of main effects that enable the integration of AMMI1 and AMMI2 biplots. Although there are some other solutions [19], the use of color scale proved to work well if a large number of genotypes need to be plotted.

*4.4. Consequences for a Breeding Program*

The most plausible explanation for the differences in adaptability between two RIL populations could be attributed to the use of widely adapted cultivar Bezostaya-1 in one cross (Bezostaya-1 × Klara) as opposed to the use of two cultivars from the same breeding program in another cross (Monika × Golubica) which were never grown on a large scale in the region of origin. It fits well the expected outcome and confirms the soundness of the standard crossing approach/strategy where one of the parents used for crossing should be widely adapted cultivar for the trait of interest. Furthermore, BK cross produced much more transgressive segregants, thus providing a wider base for selection of RILs with broad or specific adaptation in earlier generations. AMMI analysis provides the means for an easy identification of the potentially interesting RILs by simple visual inspection of biplots. The selected RILs have promising potential for use in the breeding programs aimed at quality improvement.

**Supplementary Materials:** The following are available online at https://www.mdpi.com/article/10.3390/agronomy11061022/s1, Figure S1: Pearson's correlation coefficients across environments for BK population, Figure S2: Pearson's correlation coefficients across environments for MG population, Figure S3: Modified AMMI2 biplots for (a) WGC, (b) MTW, and (c) MPH traits of BK RIL population, Figure S4: Modified AMMI2 biplots for (a) GPC, (b) MPT, and (c) MTI traits of MG RIL population, Table S1: Average daily temperatures (°C) per location during three growing seasons, Table S2: Rainfalls (mm) per location during three growing seasons, Table S3: Soil temperatures (°C) at 5 cm soil depth per location during three growing seasons, Table S4: Structure of nested effects in optimal models for each trait in both RIL populations examined, Table S5: Summary of parental means, RIL means and ranges, and rates of transgressive segregants within environments for seven quality traits assessed in BK RIL wheat population, Table S6: Summary of parental means, RIL means and ranges, and rates of transgressive segregants within environments for seven quality traits assessed in MG RIL wheat population, Table S7: Results of tests for IPCA terms for four mixograph traits containing missing values in the BK population, Table S8: Results of tests for IPCA terms for four mixograph traits containing missing values in the MG population, Table S9: "Winner" RILs within each environment for both populations examined.

**Author Contributions:** Conceptualization, I.P., J.G., and D.N.; methodology, R.Š.; formal analysis, I.P. and J.G.; investigation, R.Š.; writing—original draft preparation, I.P. and J.G.; writing—review and editing, J.G. and D.N.; visualization, I.P.; supervision, J.G. and D.N. All authors have read and agreed to the published version of the manuscript.

**Funding:** This study has been fully supported by the project KK.01.1.1.01.0005 Biodiversity and Molecular Plant Breeding, Centre of Excellence for Biodiversity and Molecular Plant Breeding (CoE CroP-BioDiv), Zagreb, Croatia.

**Institutional Review Board Statement:** Not applicable.

**Informed Consent Statement:** Not applicable.

**Data Availability Statement:** The data were obtained from the Agricultural Institute Osijek and are available on request from the corresponding author with the permission of the Agricultural Institute Osijek.

**Conflicts of Interest:** The authors declare no conflict of interest.

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
