# Peer review of "Capturing GEI Patterns for Quality Traits in Biparental Wheat Populations"

_agronomy, doi:10.3390/agronomy11061022_

Round 1
Reviewer 1 Report
The manuscript describes an interesting research on the adaptability of the wheat quality traits for RIL line to environmental conditions. Study is important especially for cereal breeders. It would be worth adding a winning list of the RIL lines, especially those with a wide adaptation (e.g. in a supplement). The following presents its detailed comments:
Line 124-125.What were the weather and soil conditions in these environments.
line 137. Many abbreviations are not explained when first used eg TW
line 162-171. There would be a need to more formalize the model used (e.g. giving an equation), and the variance and covariance structures used (if used). I don't understand what was the need to exclude certain effects in models. Rather, it looked at the following studies:
Buntaran, H., Piepho, H.‐P., Hagman, J. and Forkman, J. (2019), A Cross‐Validation of Statistical Models for Zoned‐Based Prediction in Cultivar Testing. Crop Science, 59: 1544-1553. https://doi.org/10.2135/cropsci2018.10.0642
line 174 - 177. Were they any type of adjusted means (BLUP means or LS means) using the model from the first stage?
line 200-202.
The presentation of these packages is unnecessary. Rather, they are technical issues related to the manipulation of data sets.
Discussion section. It would be useful to add information as perhaps the reason for the differences in adaptability between the studied two RIL populations.
Reviewer 2 Report
This manuscript aims to investigate the genotype by environment interactions in end use quality traits for two populations of wheat across six environments. The authors demonstrate that environment is the dominant source of variation for grain protein content, wet gluten content and test weight, but that GxE has a more important role on mixograph traits. Using AMMI biplots they identify some superior, broadly adapted genotypes. However, there is little in the way of a conclusion presented. This reviewer would like to see some suggestion of broader implications, recommendations for making selections in a breeding program, or a next step in working towards those. It is not clear that there is anything revealed in this AMMI analysis that wouldn’t be clear from a traditional breeder’s selection strategy.
Writing quality is fine, but could be improved by editing to be more direct and less wordy. A list of abbreviations would be very helpful and I think should be added as a footnote to the Tables and possibly to the caption of Figure 1.
There is some information missing in the material and methods. There needs to be more information on the growing conditions and trail set up: plot size, fertility rates etc.
The authors present some discussion on the large effect of E on GPC and WGC as being unusual in studies using RILs, however of course a large E effect can be exerted on GPC with varying fertility levels and high/low yielding conditions.
This paper is focused on quality traits which are of course important in wheat, however, is it appropriate to leave yield out of the conversation entirely? GPC and yield are antagonistic traits and the trade off is important. A stable high quality line with very low yield potential is of little interest.
Minor errors noticed:
Line 83 – typo: …Recombinant Inbred (Inbreed) Lines (RILs)…
Line 139 states that TW is expressed in kg/ha (hectare) which is a typical unit for grain yield. This should be kg/hl (hectolitre)
Line 224 – “3.2 Transgression in Quality Traits” – this is not a common use of the term Transgression – I’d stick to “Transgressive segregation in Quality Traits”
Round 2
Reviewer 1 Report
The manuscript has been significantly supplemented and revised
Reviewer 2 Report
The manuscript has been substantially improved by the revisions made by the authors. While I still maintain that yield data, even with these small plot sizes and low number of replications, would be useful to put quality traits in context. I recognize that this is at least partially my bias as a plant breeder and that this experiment was not designed to collect grain yield data. The lack of yield data does not invalidate any of the data presented. I find that this manuscript is ready for acceptance.
I have one very minor suggestion - replacing "much" with "many" in line 560 -561 so it will read "Furthermore, BK cross produced many more transgressive segregants,..."